# Smart Shoe Insole Based on Polydimethylsiloxane Composite Capacitive Sensors

**DOI:** 10.3390/s23031298

**Published:** 2023-01-23

**Authors:** Francisco Luna-Perejón, Blas Salvador-Domínguez, Fernando Perez-Peña, José María Rodríguez Corral, Elena Escobar-Linero, Arturo Morgado-Estévez

**Affiliations:** 1E.T.S. Ingeniería Informática, Avda. Reina Mercedes s/n, Universidad de Sevilla, 41012 Seville, Spain; eescobar@us.es; 2Department of Automation, Electronics and Computer Architecture and Networks, Escuela Superior de Ingeniería, Universidad de Cádiz, Avda. Universidad de Cádiz 10, 11519 Puerto Real, Spain; blas.salvador@gm.uca.es (B.S.-D.); fernandoperez.pena@uca.es (F.P.-P.); josemaria.rodriguez@uca.es (J.M.R.C.); arturo.morgado@uca.es (A.M.-E.); 3Department of Computer Science and Engineering, Escuela Superior de Ingeniería, Universidad de Cádiz, Avda. Universidad de Cádiz 10, 11519 Puerto Real, Spain

**Keywords:** smart insole, deep learning, capacitive sensor, PDMS

## Abstract

Nowadays, the study of the gait by analyzing the distribution of plantar pressure is a well-established technique. The use of intelligent insoles allows real-time monitoring of the user. Thus, collecting and analyzing information is a more accurate process than consultations in so-called gait laboratories. Most of the previous published studies consider the composition and operation of these insoles based on resistive sensors. However, the use of capacitive sensors could provide better results, in terms of linear behavior under the pressure exerted. This behavior depends on the properties of the dielectric used. In this work, the design and implementation of an intelligent plantar insole composed of capacitive sensors is proposed. The dielectric used is a polydimethylsiloxane (PDMS)-based composition. The sensorized plantar insole developed achieves its purpose as a tool for collecting pressure in different areas of the sole of the foot. The fundamentals and details of the composition, manufacture, and implementation of the insole and the system used to collect data, as well as the data samples, are shown. Finally, a comparison of the behavior of both insoles, resistive and capacitive sensor-equipped, is made. The prototype presented lays the foundation for the development of a tool to support the diagnosis of gait abnormalities.

## 1. Introduction

The biomechanical analysis of footfall and the gait is research field that is attracting more attention nowadays. It is recognized as a very effective tool for dealing with various issues associated with the march. On the one hand, these tools can be found in sports for performance analysis and feature optimization [1,2,3]. On the other hand, on the clinical side, it is used to diagnose pathologies and physical or behavioral abnormalities [4,5], as well as to monitor rehabilitation and rectification processes [6,7].

The lower limbs are the ones that suffer more during the movement of a person. Thus, they are prone to numerous diseases. With the increase in life expectancy and the ageing of the population, there are increasing numbers of cases of pain and pathologies that affect these parts of the musculoskeletal system [8]. However, the appearance of these problems is not only concentrated in the elderly population. According to [9], around 24% of people over 45 years report frequent foot pain. Many of these disorders arise as a result of having some physical abnormality not treated efficiently, such as flat feet, which gradually affects the rest of the musculoskeletal system. Another possible cause of the appearance of chronic pathologies of this nature is associated with daily bad walking habits [10,11,12]. Some of these bad habits are moving with an unbalanced load with respect to the center of the spine, or having a very marked pronator or supinator gait, which deviates from safety margins, which can cause metatarsalgia [13]. All of these anomalies and bad habits require analysis to identify them and the design of an action plan to solve them. Furthermore, the progress of this action plan has to be analyzed over time.

Gait abnormalities associated with specific clinical cases, such as the appearance of tumors, Parkinson’s disease, or cerebral palsy, are also common, according to [14]. In order to prevent severe damage, each of these cases should be analyzed and monitored, over time, specifically. The tracking of each case separately is a complex task.

The methods to analyze the gait include visual observation. This has a limited precision and can change between individual medical professionals (clinician). Therefore, visual observation is often complemented by standardized tests [15], which can be performed with different measurement equipment. The most frequent electronic measurement resources are located in specialized rooms, called Gait Labs [16]. The measurement systems of these rooms include cameras and sensor platforms that capture the trajectory and distribution of pressure on the sole of the foot. The information collected is then analyzed to identify patterns and calculate important characteristics in the clinic. Although these types of instruments are generally accurate, they cause physical and cognitive conditioning in the patient during the collection of the data, as they focus on performing tests, for which there may be bias in the measurement [17]. Furthermore, creating this type of room is economically expensive, and the information collected is limited to the time of the testing session [18].

The latest alternatives to the previously described approach are focused on the integration of measurement sensors into wearable devices. These devices can record or transmit data while the user performs daily activities [19]. This approach allows continuous analysis while distancing the user from the data-collection process. The method to collect the maximum amount of data of the footprint, static or while walking, is by recording the pressure distribution of the different regions of the foot. There are several studios and commercial solutions which have created specialized footwear for this purpose [20]. Resistive sensors are widely used because of their low cost and ease of implementation. However, they have a very abrupt response when exposed to pressure, which makes it difficult to extract precise information from the different regions of the foot [21]. Another alternative is the use of capacitive sensors. The effectiveness of this alternative depends largely on the type of materials used to fabricate the sensors. The materials must be plastic enough to allow an easy deformation but, at the same time, have the ability to return to their original decompressed state. Another requirement to address is a linear behavior between the deformation and reversion to the original shape. This is needed to achieve pressure measurements with higher resolution.

Another important feature in this kind of footwear is the distribution of the sensors in the insole. Simple solutions place sensors in the areas where most pressure is usually exerted, that is, in the area where metatarsals and phalanges join the foot, in the most distal area, and in the heel [22,23]. However, this may not be sufficient to analyze pathological cases. Adopting a contrasting perspective, some designs use a mesh of sensors which are evenly distributed throughout the insole [24,25]. These solutions achieve higher resolutions. However, the device is more complex, since it has to provide electronic resources and computational logic capable of managing the connection and reading the data from each sensor. This complexity affects the collection times, the energy autonomy time, and the device cost. A compromise solution may be the most advantageous: distributing a series of sensors in regions where pressure is usually exerted but also covering, to a lesser extent, regions of the sole of the foot where applying more pressure than appropriate reveals pathological cases or anomalies.

This paper presents the design and characterization of insoles to measure plantar pressure composed of 12 capacitive sensors for each foot. The novelties of the manuscript are: (1) The poly(dimethylsiloxane) (PDMS) material used as dielectric with an additional shock-absorbing role. The features of the material allow reversible deformation when subjected to pressure, providing high resolution in terms of data collection and continuous use without breaking or yielding the insole. (2) The smart distribution of the sensors. They are placed to identify the pressure exerted in regions where normal and abnormal patterns can be identified, such as in the case of flat feet, hypersupination, or hyperpronation. (3) The reading of the sensors is carried out by means of a single integrated circuit for each insole, simplifying and reducing the time for the acquisition of measurements. (4) A complete process of the elaboration of the capacitive plantar insole is outlined; this is an alternative design to other proposals which does not require complex or expensive laboratory tools for its development.

The rest of the manuscript is organized as follows. Section 2 summarizes recent works that propose alternative capacitive insoles, as well as designs that use PDMS as a component in plantar insoles. Section 3 describes the materials used and the design of the insole, as well as the design of the system as a whole and the methodology used to characterize and analyze the system. In Section 3, the results and discussion are shown. Finally, Section 4 presents the conclusions.

## 2. Related Works

In the current state-of-the-art, there are already proposals for wearable systems with capacitive insoles. Commercial solutions such as Motion SCIENCE [26,27,28], for which manufacturing processes are not available for replication, are common. Other studies, less scarce, do propose their own insole design with capacitive sensors.

In the study of [29], capacitive sensors composed of a glass epoxy PCB as a conductive layer and rubber as a dielectric are proposed. Four sensors were attached to a rubber sole in its lower part to make the system. The measurement system is completed with a simple CDC of its own design, an ATMEGA8 microcontroller, and an XBEE module to send the results. The load results show a response close to linear, although with a progressive decrease in response to loads greater than 40 kg over the entire area of the insole. The rubber material used to manufacture the sensor is not specified.

Similarly to the previous study, in [30], sensors which include rubber as a dielectric layer are proposed; however, a conductive textile “W- 290-PCN” from the manufacturer A-Jin is used. The method of integrating the sensors in the plantar insole is more sophisticated: a rubber insole with grooves is made to house the conductive layer and the cables that allow the connection with the acquisition system. In this proposal, there are 10 sensors which are integrated into the insole and have to be placed in more peripheral areas because the central area is reserved for locating the connection cables for each sensor. A commercial CDC MPR121QR2, an STM32F103 microcontroller, and a bluetooth module are used as the main components of the collection and transmission system. Data on the response to different loads are not provided, but a comparison of the response to walking is made between the proposed system and another commercial F-Scan system, obtaining similar results. In this study, the composition of the gum used is also not provided.

In the work [31], an insole is presented whose sensors use a double layer of copper and an EMFIT electroactive ferroelectric film as an intermediate layer. There are eight sensors located on the insole, in positions of greatest interest based on the pressure zones of the foot during walking and treading. In the version of the referenced study, the fragments of EMFIT are glued together with the copper inside two layers of copper that cover the foot area. In a later, more sophisticated version presented in [32], EVA rubber is used to adhere the sensors. The second publication presents the hardware used, consisting of FDC1004 as CDC, an LPC824 microcontroller, and a bluetooth module SPBT3.0DP2; this system is integrated into the insole itself in the area of the plantar vault. The study shows that the response is linear from loads of 600 KPa.

The study presented in [21] proposes the design of a low-cost insole, in which the sensors are made up of silver cloth as a conductor and four layers of cotton cloth. The insole consists of two layers of cotton that contain three circular-shaped sensors, which have a very high radius with which they cover the entire area of the foot. As elements of the acquisition system, there is a PCAP01 as CDC, a PIC18F25K80-I/MM microcontroller and a bluetooth module. Although it does not present a linear response of pressure with respect to capacitance, the study reveals that the derivative with respect to an initial capacitance is close to linear.

The study [33] presents a sophisticated design composed of small triangular links, each of which has twelve capacitive sensors made of copper and a silicone rubber foam to which a spray of electrically conductive silicon rubber is applied. Each of the triangles records the measurements of an AD7147 CDC controller. The measurements are sent and processed by a single controller, and sent by I2C. The performance of the system is accurate given the number of pressure sensors, a total of 280; however, the complexity of the system requires special shoes that limit portability.

This work proposes the use of PDMS as a flexible dielectric material for the development of a capacitive insole. In previous work, the plantar insole proposals use PDMS as a cover for piezo-resistive sensors. In [34], a simple design is presented that does not require complex tools for its elaboration or welding process, but no acquisition system is presented and the difficulty of connecting the outputs, which are distributed along one edge of the insole, is not resolved. In [35], PDMS is also used, creating a composition with multiwalled carbon nanotube (MWCNT) to make a piezo-resistive sensor; the insole design in this case is complex, requiring sophisticated printing machines. In this study, no complete acquisition system is presented either. In work [36], the authosr again use a composition of PDMS and MWCNT to cover the electrodes, creating piezoresistive sensors. A plantar insole of seven sensors is proposed, which is arranged on a layer of PET substrate, and a mold is used to create the layer of a PDMS-MWCNT composite in the place where the electrodes are located. The manufacturing process of the PET substrate layer with the electrodes is not detailed.

The proposal of this work consists of a complete system in which PDMS is used as a dielectric intermediate layer and copper is used as a conductor to develop capacitive sensors. PDMS is not only distributed in the location of the sensors but also covers the rest of the area of the plantar insole, so it has the same composition and thickness, improving ergonomics, and without conditioning the distribution of pressure during footsteps due to the presence of different materials on the surface of the insole. The way to manufacture this layer is from a mold, which can be customized without the need to alter the manufacturing process, providing the possibility of creating different plantar-insole sizes, and changing the size, shape, and distribution of the sensors.

Second, building on the insoles developed in this work, future work is intended to redirect the studies of footprint analysis begun by [37,38], in which machine learning is used to identify gait pathologies with resistive insoles. With current improvements in hardware performance and communication technologies, advances in the development of wearable IoT devices for health monitoring have been increasing. In the context
of body mechanics analysis, this includes applications ranging from stress detection [39] to identifying biomechanical habits that pose a risk to health [40]. With the use of capacitive insoles and the use of PDMS as a dielectric, given its mechanical qualities, it is hoped to achieve better resolution in the data than those obtained with resistive sensors, allowing one to take advantage of the potential of advanced machine-learning algorithms, which can identify subtle features and are tolerant to noise.

## 3. Materials and Methods

### 3.1. Capacitive Insole Composition and Materials

Each of the sensors in the insole consists of two electrodes separated by an insulating dielectric. This dielectric must be flexible enough to permit the pressure of the foot to cause a deformation in the dielectric, which eventually changes the distance between the two electrodes. The capacitance variation with respect to distance can be computed using Equation (Equation 1).

(1)
$$C=\frac{\epsilon_{r}\epsilon_{0}A}{d}$$

where *C* is the capacitance of the capacitor (F), 
$\epsilon_{r}$
 is the dielectric constant (2.3–2.8 for PDMS [41]), 
$\epsilon_{0}$
 is the vacuum permittivity (8.854 × 10^−12^ F/m), *A* is the area (m^2^) and *d* is the distance between plates (m).

The plantar insole proposed in this paper relates the distance between several electrodes with the ground. Thus, an intermediate material must be included. For this reason, the capacitive insole designed is composed of several layers (see Figure 1): (1) the first layer is the ground of each of the sensors which is common to all of them; (2) a flexible and thick-enough second layer with the insulating element which acts as a dielectric and (3) the last layer with all the measurement electrodes placed at specific strategic points.

#### PDM-Based Dielectric Layer

The three materials most commonly used in the literature as dielectrics in capacitive insoles are: PDMS, PVC and Ecoflex. PDMS is the material with the best pressure measurement range, the best sensitivity and the best durability [42,43]. PDMS is a type of silicon-based material which is transparent. Its mechanical properties depend on its fabrication process. Its viscoelastic behavior is the reason why it has been frequently used in applications such as the one presented in this paper. Its properties as a dielectric make it an excellent choice for the purpose of this insole design [44].

The thickness of the insole affects not only the sensor sensitivity but also the comfort of the device. In order to select the best thickness, different tests were performed with a single sensor. For the dielectric, several PDMS cylinders with a radius of 15 mm and different heights (2, 3, 4 and 5 mm) were made. Cylinders were placed between two copper electrodes of the same radius to create the capacitive pressure sensors. Since the size of these cylinders is small, the pressure performance study was performed using a screw and connecting the electrodes to a development board which includes a CDC AC7146. With each 1/4 turn of the screw, the pressure on the device increased by 17 KPa, reaching a final pressure of 119 KPa. After several pressure tests, the sensitivity results can be seen in Table 1. To calculate the sensitivity, the applied pressure is related to the measured capacitance; the slope of this curve provides the sensor sensitivity.

Only the 2 mm sensor broke under a pressure of 85 KPa because of its thickness. Therefore, it was decided to use the 3 mm sensor which has a very similar sensitivity to the previous one and avoids the breakdown of the device. For this reason, a thickness of 3 mm was selected. It is considered sufficiently comfortable to be used as a insole and it provides an adequate sensitivity.

### 3.2. Sensors Distribution

The location of the sensors is designed to cover the entire sole of the foot but specifically consider the areas that provide more information. According to [45], the areas of the sole of feet that provide more information are those where the metatarsal bones and the proximal phalanges are located. Another area where a wider pressure is exerted is where the calcaneus and the talus bone sit, which mainly make up the heel. By analyzing the load distribution in these areas, medical professionals can identify and monitor pathologies such as Hallux abducto valgus (HAV) deformities [46] or hammertoes [47]. The remaining covered areas correspond to the area where the medial and distal phalanges rest and where the so-called plantar arch or vault is located. The information collected in these areas complements the information required to analyze common pathologies such as pronation and supination. Furthermore, a lower or higher pressure identified in the area occupied by the plantar vault is associated with deformities such as pes cavus or flat feet [48,49].

Considering the works mentioned, the final location of the sensors in the insole is shown in Figure 2. Twelve sensors are used, five located in the metatarsal and proximal phalanges area, two in the heel area, two in the medial and distal phalanges area and three in the area of the plantar vault. This distribution is similar to the one considered in previous works [31,50].

It is important to note that to ensure that the performance of the sensors is not affected by changes in the curvature of the plantar insole, in the design of the dielectric layer we separated the part where each sensor is located with a circular groove, 3mm thick in its circumference.

### 3.3. System Design

The system to obtain the samples, from the developed capacitive insole designed, is shown in Figure 3. The developed insole is connected to an AD7147-1 controller, a programmable chip equipped with a capacitance to digital converter (CDC) and an on-chip environmental calibration. This chip has an interface compatible with 
$I^{2}C$
, which is used to set the configuration and to manage the system through an ultra-low power microcontroller (STM32L432KC) of the STM32L series with Arm Cortex-M4 MCU. The system also includes a bluetooth module which allows the user to transmit the data collected and is managed by the microcontroller and battery-power outlets.

Additionally, to record and further analyze the data collected and sent by the system during the study, a support application was developed. The following sections explain, in detail, the features and tasks performed by each component of the system.

#### 3.3.1. Embedded Controller System

The read process performed by the AD7147-1 controller is designed to iterate through a series of sequential steps, which are called stages. It can be configured to up to 12 stages. Each stage can be custom configured to measure the capacitance between two sensors or between a sensor and the ground layer. For our purpose, twelve stages were configured. In each of them, a capacitance reading of each sensor is obtained with respect to the common ground layer of the plantar insole. The value obtained at each stage is stored in a specific stage register. At the end of each iteration, i.e., once all the configured readings have been carried out, the AD7147 notifies the reading iteration of this completion through interrupts.

The STM32L432KC microcontroller is the component that performs the stage configuration of the integrated CDC controller. It is also in charge of accessing the internal registers of the AD7147-1 to obtain samples read from the plantar insole. The reading process is performed once all the reading stages are completed by the CDC controller.

#### 3.3.2. Application

The developed application receives the data recorded periodically by the measurement system through bluetooth. The values obtained from each of the sensors included in the insole are shown in streaming using a graphical interface. Additionally, at the request of the user, the application can store values taken from each sensor individually, calculated from the average of a sample set, in order to eliminate fluctuations. This functionality enables the collection of data in specific tests of this work whose objective is to analyze the behavior of the sensors when subjected to a certain weight, with different plantar postures or gait stages.

### 3.4. Capacitive Plantar Insole Analysis

The main objective of this work is to design a system that allows high precision in the measurements recorded during walking while using the capacitive sensors insole. A comparison of the response of the plantar insole designed in this study with an insole whose sensors are resistive, specifically force-sensitive sensors FSR^TM^ 400 sensors, is made to check if the system presented achieves higher resolution. The FSR plantar insole used for the comparison was developed and analyzed in [37,38]. It was designed to identify common gait abnormalities, i.e., pronation and supination, and some common anomalies to a degree that may be dangerous to the user or pathological. In contrast, in this paper, the objective is to design a system which supports the identification of a greater variety of pathologies.

To analyze the behavior of the plantar insole with respect to the force exerted under controlled conditions, a set of calibrated weights was used. The weights were placed on top of a base, weighing approximately 1.45 g, which has an area with the same diameter of each sensor included in the insole. In this manner, the total weight of each calibrated weight is concentrated in the entire area of each sensor. This process is used to both check that a higher resolution response is achieved when the weight is exerted, thanks to the use of the PDMS-based capacitive sensors, and obtain values that allow the individual calibration of each one of the sensors included in the insole.

The analysis of the system continues with the assessment of the response when a user is already using it to register the march. For this, a capacitive and resistive insole of each size were used. The user wearing the insole was asked to walk at a calm pace while the values acquired by the system were recorded. Observation under these conditions allowed the identification of limitations of each insole in a context closer to a real situation.

## 4. Results and Discussion

### 4.1. System Development

In this section, the prototype development process is described. Details of the implementation of the capacitive plantar and acshield insole are given. The latter component serves as a motherboard to connect the elements that make up the complete measurement system.

#### 4.1.1. Fabrication Process of the Plantar Insole

In order to fabricate the PDMS layer, a methacrylate mold was made with the desired shape. Then, the PDMS layer was poured for curing. A laser cutter (Epilog Laser Helix 50 W) was used to cut the methacrylate pieces. All mold pieces were obtained by cutting 150 × 300 mm methacrylate sheets. The mold was made up of a base, a border to prevent PDMS leakage and twelve concentric circles (with radius of 9.5 and 7.5 mm and height of 1 mm) placed where the measurement electrodes were superimposed. This gives the shape of the capacitive pressure sensor in these areas of the insole.

The fabrication process of the PDMS needed 50 g of PDMS with 5 g of curing agent (since the ratio should be 10:1). The mixture was degassed in a vacuum chamber and poured into the insole mold. The cure was carried out in an oven, previously preheated to the same temperature, for one hour at 80 °C. Once it was cured, the PDMS could be carefully removed from the mold using a flat spatula to avoid breakage.

The fabrication process of the measurement electrodes needed flexible PCBs which consist of 35 μm copper on a Kapton substrate (AN10, CIF). The first step was to clean with acetone and isopropanol. After drying, photoresist (Positiv20, Kontatkt Chemie) was deposited on the copper layer by spraying at 30 cm and at an angle of 
$45^{\circ }$
. The photoresist was cured in the oven for 15 min at 80 °C. An acetate-based photolith film was used to print the shape shown in Figure 2b. This shape included the electrodes, tracks and ground fields that could be connected to the ACShield pin. This film was placed over the photoresist. A blackening spray (Black Covering, Abezeta) was applied to the photolith to prevent the penetration of UV light. With the photolith film aligned on the photoresist, they were placed in the UV lamp (300-245, LPKF) for 90 s.

The development of the photoresist was carried out by placing it for 3 min in a tray with 3.5 g of caustic soda dissolved in 500 mL of water and gently agitated. For chemical etching, a solution of 110 vol. hydrogen peroxide (20%), hydrochloric acid (20%) and water (60%) was used. The flexible PCB was introduced into the solution until the undesired copper was removed. Electrical continuity tests were performed to ensure that there were no defects. Finally, it was cut to the shape of the insole. The result of the different components can be seen in Figure 4.

Compared to other recent studies presenting a full capacitive template design, the described manufacturing processes require the use of more complex tools, with the exception of [32], which does not solve the problem of connecting the insole to a recording system, and [34], which has a simpler design and low life cycle due to material used. Additionally, the more complex tools used in the manufacturing process can be substituted by others and their use may not even be necessary. The mold can be made with materials other than those used in this study. On the other hand, PDMS can be solidified without the use of an oven, at room temperature, but the process will take longer. In the same way, the use of a UV lamp is not necessary.

Finally, the ground layer consists of a copper foil bonded to a Kapton film. The copper foil was cut using a laser labeling machine (EASY 200F, SISMA). Then, the copper foil was adhered to Kapton tape, which acts as an electrical insulator and mechanical bonding. The copper foil was obtained from a 50 mm thick copper tape. This was the reason why two pieces of tape were used to make the ground layer of the insole. Both halves were electrically connected using two tin solder joints.

All of the layers described above were aligned and taped together with Kapton tape to ensure electrical insulation and fixing. Kapton is an insulating material which is known for its recurrent use in electronics because of its flexibility, resistance, and dielectric properties. Additionally, the ground layer was connected to the flexible PCB layer using the track designed for its connection to AD7147. Finally, for better performance, the electrodes were protected with a copper field around them, which was connected to the CDC controller.

The insoles were made for both feet with sizes that are close to the average for men and women. Specifically, sizes 43EU (27 cm long) and 39EU (24 cm long) were fabricated.

#### 4.1.2. Connectivity of the System

A PCB shield was used to allocate the AD7147-1 CDC controller and connect it to all of the system components. These components are: an STM32 NUCLEO-L432C MCU, a bluetooth module, two 7.4 V LiPo batteries and the capacitive plantar insole. The connectivity details can be found in Table 2. The copper shield built around the electrodes is connected to the ACShield pin of the AD7147 integrated circuit.

### 4.2. Insole Response Analysis

#### 4.2.1. Calibrated Weights Response

The behavior analysis phase of the insole was performed using a set of calibrated weights of 20 g, 50 g, 100 g, 200 g, 500 g, 1 kg and 2 kg. The behavior of each capacitive sensor without the rest of the insole was checked using a circular base of the same diameter as the electrodes and placing each weight on top of it. The weight of the base is about 1.45 g (measured with a precision scale). The use of combinations of stacked weights to obtain more response values was avoided due to the flexible feature of the insole added to the small area of the base; it resulted in an unstable situation in the entire stack.

Table 3 shows the values obtained by exercising the different weights in the EU38 size insole, specifically, the right foot insole. Five repetitions were performed for each calibrated weight under the same environmental conditions. In order to achieve the same conditions for each repetition, the system was switched off and then on at the beginning of every test round.

In Table 3, it can be seen that the increase in capacitance value is directly proportional to the weight exerted. In some cases, particularly for this EU38 capacitive insole and the right foot, abnormal values were obtained. These outliers are associated with minor changes in the electric field of the environment close to the insole or with automatic calibration failures caused by the CDC Controller. However, these errors did not occur in subsequent measures. In any case, this event is reflected in the transparency and repeatability of the study. The CDC provides a 16-bit integer value resolution. The values are then converted to capacitance units according to the datasheet of the manufacturer.

Table 4 shows the results obtained when different weights are exerted on a capacitive EU 43 insole size. The process followed to obtain the samples is explained in Section 4.2.1. All the values obtained show the directly proportional behavior of the capacitance with respect to the exerted weight; anomalous data were not obtained.

As happened with the EU 38 size capacitive insole, it can be seen that the base capacitance value of each sensor is different. Three features can be mentioned to explain these differences: (1) small variations in the capacitive insole thickness, (2) the arrangement of each electrode on the flexible PCB with respect to the rest of the sensor tracks in the insole, and (3) the degree of coverage of each track by the copper plane that covers them. However, this change in the coordinates of the origin does not imply a problem in the measurement, since the values can be subtracted or normalized with low computational cost.

The comparison was made between the developed capacitive insole and the one previously introduced, which includes FSR sensors. The test conditions were the same as in the capacitive insole: the same calibrated weights and base platform were used. It is important to note that the base platform is the same despite the fact that the diameter of the FSRs that make up the resistive insole is smaller (approximately 7.62 mm). The results can be seen in Table 5.

As can be seen in Table 5, the FSR sensors do not detect any change in weight up to 200 g. A response appears when a weight of 500 g is exerted on the insole. The resistive insole used in this study uses voltage dividers to acquire the values. The resolution of the ADC of the microcontroller is 12 bits and the power supply given to the sensors is 3.3 V. Table 5 shows the reverse voltage values obtained after reading the sensor, that is, 3.3 V 
$-V_{out}$
. This inversion is performed for the sake of clarity in terms of visual comparison. It can be seen that with this simple conversion, the results obtained are also directly proportional to the weight.

Figure 5 and Figure 6 show the average results for each repetition performed using the EU 38 size insole, as well as the combined average of all repetitions. It can be observed that for small weights the response of the sensors follows a logarithmic trend. This response may be due to the fact that the upper and lower layers of the plantar insole, that is, the shield layer and the one with the electrodes, are more separated at the beginning, so that, with small loads, the mechanical properties of the PDMS do not have much impact. However, it may also be because PDMS deforms more easily up to a given weight. However, for the highest weights, from 500 g on, the behavior can be approximated to a linear function. It should be noted that the resolution of the CDC is far from its saturation value, which is around 25 pF.

Figure 5b and Figure 6b show a high standard deviation, which is almost entirely due to the change in the coordinate value at the origin which is obtained with each sensor. The proportional relationship between weight and capacitance is kept in the insole, except in anomalous cases. Looking at Figure 5a and Figure 6a, a considerable change in the coordinate at the origin for the capacitance value between the different repetitions can be observed. This issue is probably due to the the temperature change of the ACShield and the MCU in the device calibration process when the tests were performed. Nevertheless, the proportional relationship between repetitions of measurement is maintained and it becomes linear with weights starting at around 500 g. This is considered appropriate given the purpose of measuring the force exerted by the footprint in different regions of the sole of the foot. This behavior is similar to that of the study presented in [32], and is closer to linear than other studies, such as [21,31], whose trends are exponential and quadratic, respectively. With other studies it is not possible to make an appropriate comparison, because they focus on analyzing the behavior of the sensors under precise loads. The study [29] reflects a linear behavior from low weights, but its design has the disadvantage of not having the sensors integrated into the insole.

The response of the plantar insole based on FSR sensors can be seen in Figure 7. An advantage of this insole is that the values obtained with each repetition are very robust, with little variation between measurements. Furthermore, the response it provides in the range from 500 g on can be assimilated into a linear function. However, given the scope of applications for which the developed system was intended, notorious disadvantages of particular relevance are encountered. Firstly, for weights lower than 200 g, a value is not obtained. This is an important limitation since pathologies related to areas of the plantar arch could not be identifiable because they are areas in which very little weight is distributed in that area of the foot’s sole. Secondly, the slope obtained in the range of 500 g to 2 kg is very high, reaching approximately 75% of the maximum resolution of the sensor in that weight range. This is a significant limitation that can prevent the identification of activities that involve maximum force on the sole of the foot. Furthermore, it could mean that it cannot make the correct measurements in users who are overweight and prone to foot injuries and pathologies.

#### 4.2.2. Insole Response with Higher Loads

To analyze the behavior with higher loads, a screw was used. Measurements were taken from 1Kg to 8Kg. All the sensor responses were registered, but only three repetitions were carried out and using one plantar insole, specifically, the left-foot insole, size EU 43. The results obtained can be seen in Table 6. The increase in capacitance per kilogram added is small, but it follows a close-to-linear trend for all sensors. The sensitivity obtained is higher in those sensors whose bus is closest to the edge of the plantar insole. For example, the sensitivity of the s9 sensor is around 0.76 fF/KPa while that of the s5 sensor is around 0.46 fF/KPa.

Figure 8 shows the average result obtained. As indicated, the behavior of the template for these loads is close to linear. This result is consistent with the results obtained in preliminary tests of this study, and in the works [42,43]. Regarding work with capacitive plantar insoles, again, the behavior of the sensors in this study coincides with those elaborated in [29,32].

#### 4.2.3. Insole Response during Gait

In this section, the behavior of the insole is analyzed by obtaining data from users walking while wearing it. For the EU 43 capacitive and resistive insole test, the participant who wore them was a 31-year-old man weighing 91 kg. For the EU 38 size insole test, the participant was a 32-year-old woman weighing 65 kg. Participants were asked to walk at a leisurely pace while the developed registration application was used to collect the data received by each insole. For the resistive insole, two steps were carried out: (1) the data obtained by each sensor in the tests were used to scale the values and (2) a normalization of the values was carried out considering the maximum initial value of 3.3 V.

For the capacitive insoles, a calibration was also carried out from the values obtained in the tests with the calibrated weights. An additional step was added, consisting of partially screening the electric-field component exerted by the body itself in proximity to the sensors. This was carried out by dividing the sample recorded in a time instant by the mean of the values for that time instant of all sensors. Since all sensors were affected, all of them were used to perform the average.

Figure 9 shows the results for the capacitive insoles. The values were normalized for an easier comparison, so the measurements have a value between 0 and 1. The values taken for the normalization of each sensor were the maximum and minimum values obtained in the entire record, plus 10% of the difference between these two values. It should be noted that this normalization is applied for better visualization in Figure 9, although it could be a valid normalization process to use them as inputs of a machine-learning model. The results can be seen in several graphs, where the signals are grouped by region, that is, where on the sole of the foot they are located. As a result, greater activity can be observed in the sensors located in the most distal zone, in the upper metatarsus and in the phalanges. In addition, the activity of the rest of the sensors can also be seen, with a set of values in the range obtained during the previous phase. There is no loss or saturation of the values.

Figure 10 illustrates some distribution maps, where the responses of the insoles during different gait stages are shown. The results show that the insole responds to footsteps consistently. Although a more exhaustive analysis would be necessary to estimate the robustness of the system in the acquired data, the results show that it is possible to record data that can be used to train machine-learning models and develop a support system for monitoring and identifying pathologies.

The results of the gait recording test while the user was wearing the resistive insole can be seen in Figure 11. The values obtained are more revealing and easy to observe at a glance. However, as expected, at moments in which a reduced weight is exerted, the insole does not register any response. Thus, it might lose relevant information for the identification of more subtle characteristics. Additionally, it can be seen how some sensors at specific temporal instants are very close to the saturation value. It must be taken into account that the test conditions were: walking at a leisurely gait and with a participant of a high weight given his height. However, people with gait problems may have higher weights, which would make the FSR sensors reach its saturation region, thereby limiting the measurements.

These reasons are relevant enough to choose the capacitive insole presented in this paper. Its use in anomaly identification problems will allow more precise data to be obtained and, therefore, provide more information. The additional steps needed to filter out the electric field might be the downside of the processing system. However, the use of artificial-intelligence algorithms for the identification of anomalies, given the linear nature of the disturbance, is likely to cope with this type of conditioning.

#### 4.2.4. Response-Time Details

Regarding acquisition times, measurements with the highest decimation level of the CDC controler were performed every 37 s. However, the measurements were collected every 100 milliseconds using a sampling rate of 10 Hz. This rate is considered enough for gait monitoring and the identification of abnormalities. The sampling frequency could reach 25 Hz, without reducing the decimation, by reducing the data-transmission time. This reduction can be achieved by increasing the byte transmission rate, which has the disadvantage of higher consumption. Another possibility to reduce the transmission time is to compress the information to reduce the number of bytes that have to be transmitted by serial communication.

## 5. Conclusions

In this paper, a system equipped with capacitive insoles for recording anomalies in footprints was developed. To achieve the linear response of each sensor, a PDMS composition is used as the dielectric in the intermediate layer of the plantar insole. This paper shows the entire development process of the insole and the collection and delivery system. Thus, any researcher can replicate it. The proposed manufacturing process does not require the use of expensive tools. The most expensive material used is flexible PCB sheet, although this material can be replaced, for example, by using 3D printers to make flexible printed circuits.

The analysis performed reveals that capacitive sensors have advantages over resistive ones. Specifically, capacitive sensors are capable of reacting to small weights, which allows the analysis of anomalies and pathologies with more subtle characteristics. In addition, they offer a linear behavior for weights from 1Kg without saturation.

The presented design is an alternative to other capacitive insoles proposed in the literature, offering correct performance, behavior under footfall similar to studies that achieve a linear response, and totally homogeneous integration of the sensors in the plantar insole, which favors ergonomics.

The response offered by the recording system is appropriate in terms of time to identify the changes, phases, and events in the march. There is room for improvement in the transmission process and changes in decimation to perform more samples per second.

The main limitation identified in the prototype consists of the disturbance in the values due to the electric field exerted by the body. However, this disturbance can be partly removed in a simple way by performing multiplication operations. The results of the gait analysis reveal that the signal retains the gait information after the transformation. However, the search for different filtering methods could be relevant to retain information of interest for the identification of anomalies.

As a future work, the developed system will be used to record the gait of several users with pathologies or anomalies that influence gait. With the prepared data set, machine-learning models will be developed and evaluated to identify pathologies autonomously in order to develop a complete monitoring and diagnostic support system.

## Figures and Tables

**Figure 1 sensors-23-01298-f001:**
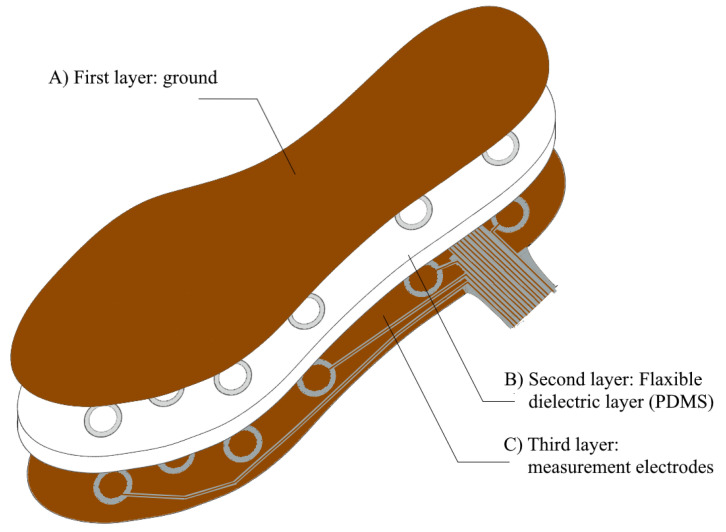
Scheme of main layers that make up the proposed plantar insole.

**Figure 2 sensors-23-01298-f002:**
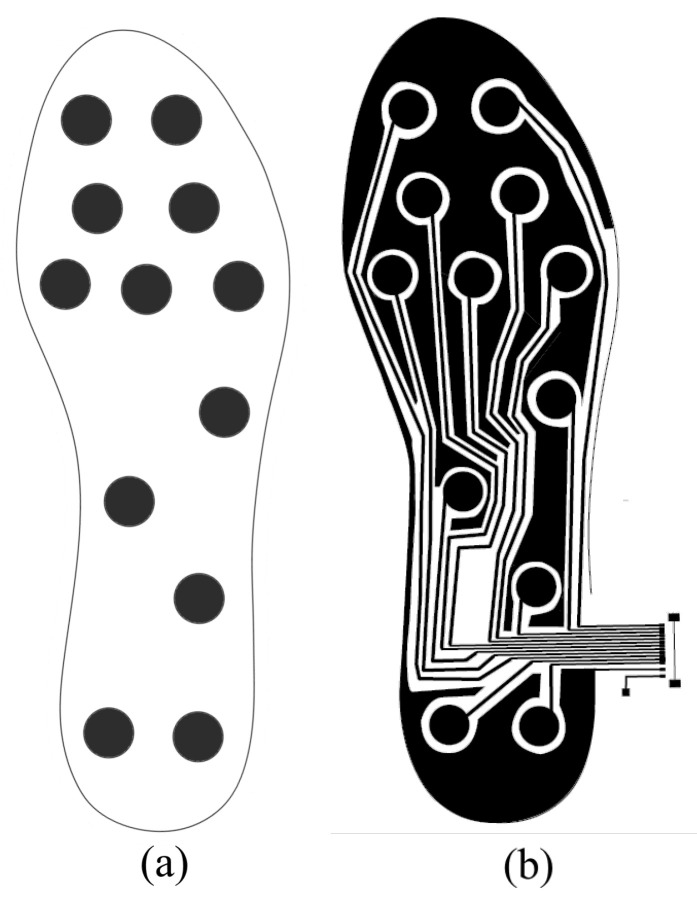
(**a**) Location of the sensors in the proposed capacitive insole. (**b**) Layout of the layer with the electrodes on the capacitive plantar insole.

**Figure 3 sensors-23-01298-f003:**
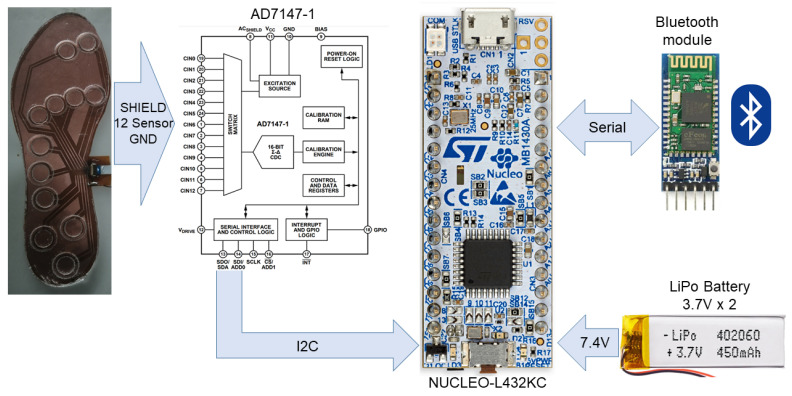
Block diagram of the main components of the plantar insole system.

**Figure 4 sensors-23-01298-f004:**
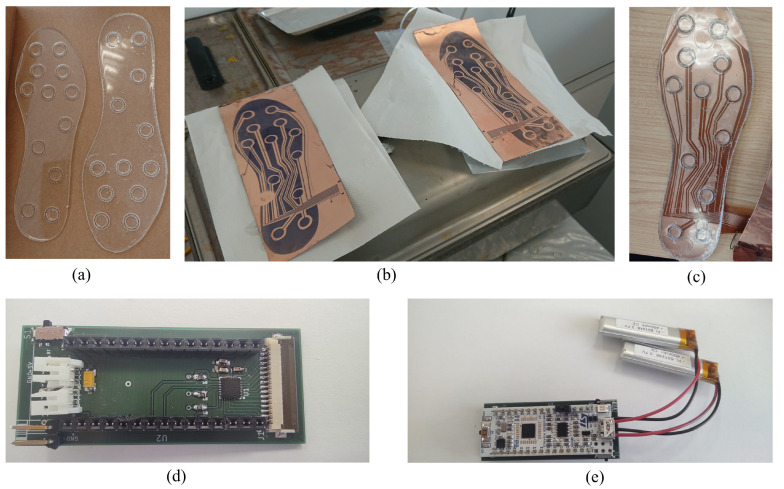
Prototype components: (**a**) PDM-based dielectric layer; (**b**) flexible PCB with remaining photoresist ink, prior to copper-dissolution step; (**c**) the insole with three layers finished, prior to fixing them with Kapton; (**d**) ACShield built to house the CDC Controller Ad7147-1 and to connect both plantar insole, batteries, bluetooth and MCU to the whole system; (**e**) ACShield with MCU and batteries connected.

**Figure 5 sensors-23-01298-f005:**
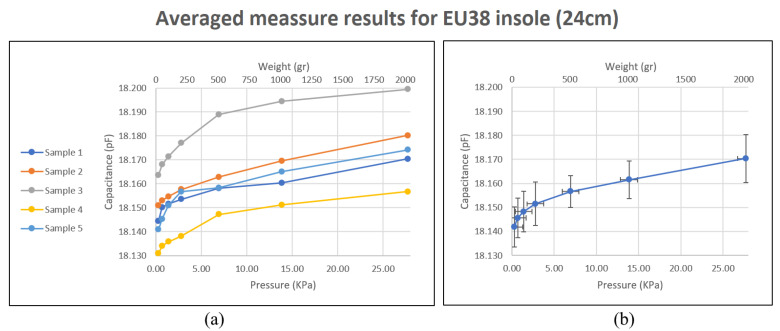
Capacitive plantar insole, size 38 (24 cm). Behavior of sensors against weight in the area they occupy. (**a**) show the average results for each repetition performed. (**b**) shows the combined average of all repetitions.

**Figure 6 sensors-23-01298-f006:**
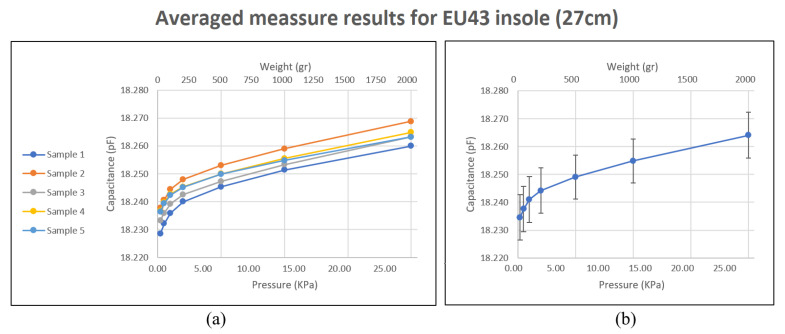
Capacitive plantar insole, size 43 (27 cm). Behavior of sensors against weight in the area they occupy. (**a**) show the average results for each repetition performed. (**b**) shows the combined average of all repetitions.

**Figure 7 sensors-23-01298-f007:**
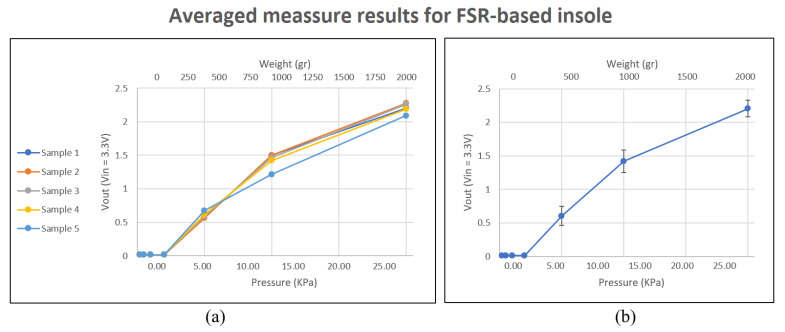
Resistive plantar insole. Behavior of sensors against weight in the area they occupy. (**a**) show the average results for each repetition performed. (**b**) shows the combined average of all repetitions.

**Figure 8 sensors-23-01298-f008:**
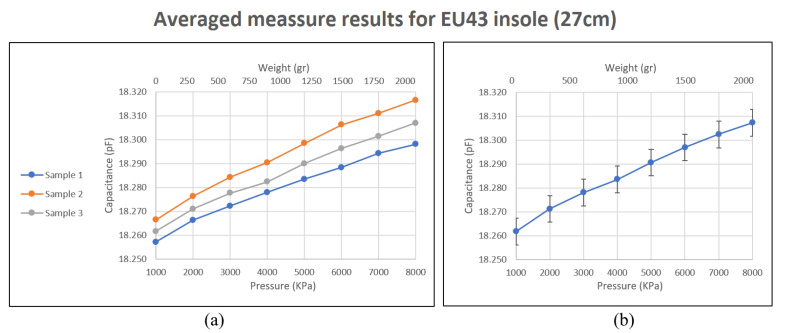
Behavior of sensors against weight in the area they occupy, Using higher loads. (**a**) show the average results for each repetition performed. (**b**) shows the combined average of all repetitions.

**Figure 9 sensors-23-01298-f009:**
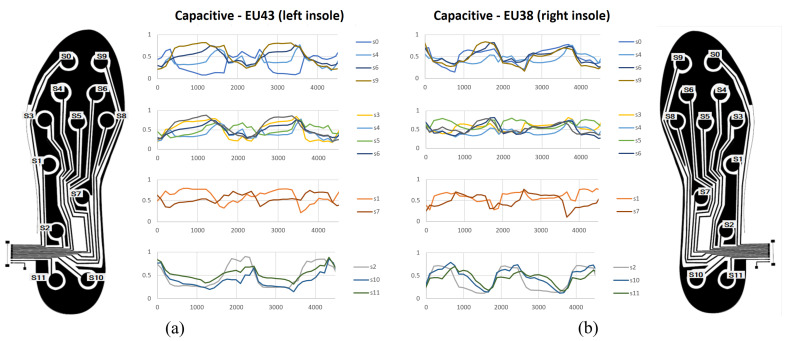
Behaviour during user gait—capacitive insoles. (**a**) EU43 size, left insole. (**b**) EU38 size, right insole.

**Figure 10 sensors-23-01298-f010:**
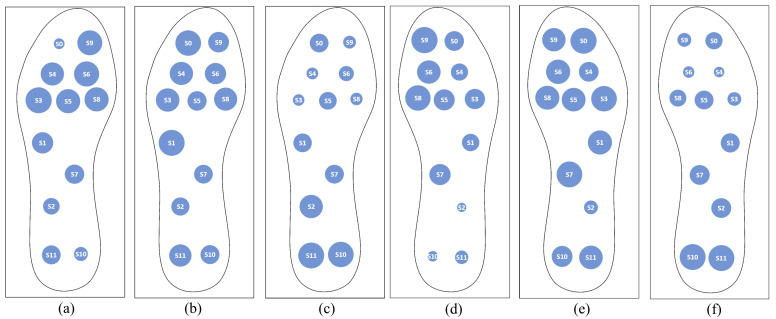
Map distribution samples obtained during different gait stages. For size EU 43 insole: (**a**) the forefoot is resting on the ground; (**b**) the whole foot is resting; (**c**) the hindfoot is resting. (**d**–**f**) reflects the same stages as (**a**–**c**), respectively, but with the size EU 38 insole.

**Figure 11 sensors-23-01298-f011:**
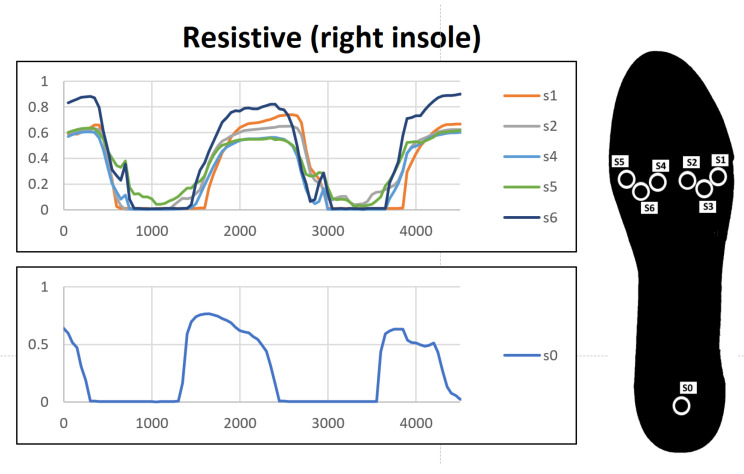
Behaviour during user gait—resistive insole.

**Table 1 sensors-23-01298-t001:** Sensitivity values and range for cylinders.

	Cylinder 1	Cylinder 2	Cylinder 3	Cylinder 4
Thickness (mm)	2	3	4	5
Medium sensivity (pF/KPa)	0.021	0.020	0.015	0.014

**Table 2 sensors-23-01298-t002:** PCB connectivity details. Each row represents a wire; if pin is written, it means that it is connected.

NUCLEO-L432C	AD7147ACPZ-1	Bluetooth Module (HC-06)	LiPo Battery 7.4 V
VIN			7.4 V
GND	GND/SDI/CS	GND	GND
5 V		5 V	
3V3	VCC/VDRIVE		
D2	/INT		
D3		TX	
D4		RX	
A4/SDA	SDO		
A5/SCL	SCLK		

**Table 3 sensors-23-01298-t003:** Plantar insole— foot size 38 (24 cm).

	Weight	S0	S1	S2	S3	S4	S5	S6	S7	S8	S9	S10	S11
R1	base	19.301	18.102	17.888	17.937	17.939	17.960	17.872	17.800	18.018	18.136	17.872	18.840
	20	19.308	18.108	17.895	17.937	17.942	17.972	17.877	17.805	18.022	18.141	17.877	18.846
	50	19.320	18.110	17.897	17.938	17.946	17.973	17.878	17.805	18.023	18.146	17.912	18.849
	100	19.323	18.112	17.898	17.940	17.947	17.975	17.880	17.805	18.024	18.150	17.913	18.849
	200	19.327	18.113	17.899	17.942	17.949	17.978	17.883	17.806	18.025	18.153	17.915	18.850
	500	19.334	18.117	17.902	17.946	17.954	17.985	17.890	17.809	18.031	18.160	17.918	18.854
	1000	19.335	18.121	17.906	17.952	17.962	17.992	17.898	17.814	18.039	18.169	17.886	18.859
	2000	19.347	18.129	17.911	17.967	17.974	18.004	17.911	17.821	18.053	18.184	17.890	18.866
R2	base	19.308	18.103	17.884	17.935	17.942	17.970	17.888	17.817	18.012	17.682	17.910	18.858
	20	19.311	18.104	17.884	17.936	17.945	17.974	17.891	17.821	18.022	17.683	17.913	18.861
	50	19.315	18.107	17.885	17.938	17.947	17.976	17.892	17.823	18.024	17.685	17.914	18.862
	100	19.317	18.108	17.886	17.940	17.948	17.978	17.895	17.826	18.027	17.688	17.914	18.863
	200	19.320	18.109	17.889	17.942	17.954	17.982	17.898	17.827	18.029	17.689	17.919	18.866
	500	19.331	18.110	17.892	17.947	17.960	17.987	17.905	17.830	18.037	17.691	17.924	18.868
	1000	19.343	18.115	17.896	17.954	17.968	17.998	17.914	17.835	18.044	17.693	17.928	18.872
	2000	19.355	18.124	17.905	17.964	17.980	18.012	17.930	17.843	18.057	17.703	17.932	18.880
R3	base	19.245	18.098	17.884	17.954	17.972	17.967	17.932	17.818	18.023	17.967	17.903	18.849
	20	19.279	18.102	17.889	17.968	17.989	18.003	17.940	17.828	18.038	17.965	17.910	18.854
	50	19.299	18.107	17.891	17.975	17.991	18.004	17.945	17.829	18.039	17.966	17.910	18.858
	100	19.316	18.111	17.891	17.980	18.000	18.001	17.942	17.831	18.042	17.974	17.913	18.859
	200	19.341	18.111	17.891	17.978	18.010	18.005	17.951	17.830	18.055	17.983	17.914	18.863
	500	19.401	18.115	17.891	17.991	18.016	18.025	17.958	17.834	18.062	17.998	17.918	18.868
	1000	19.417	18.121	17.900	17.980	18.019	18.029	17.969	17.836	18.077	18.015	17.919	18.873
	2000	19.445	18.131	17.904	17.987	18.030	18.036	17.974	17.842	18.072	18.046	17.893	18.881
R4	base	19.324	18.116	17.895	17.917	17.907	17.907	17.841	17.768	17.959	17.861	17.885	18.849
	20	19.327	18.118	17.900	17.925	17.913	17.928	17.843	17.773	17.975	17.857	17.887	18.850
	50	19.331	18.122	17.901	17.934	17.916	17.925	17.844	17.777	17.979	17.864	17.895	18.851
	100	19.342	18.117	17.903	17.937	17.922	17.928	17.842	17.777	17.977	17.859	17.896	18.852
	200	19.345	18.119	17.905	17.938	17.924	17.934	17.846	17.780	17.976	17.862	17.898	18.854
	500	19.345	18.121	17.906	17.935	17.935	18.003	17.850	17.783	17.981	17.845	17.901	18.858
	1000	19.350	18.121	17.912	17.939	17.938	18.007	17.854	17.787	17.987	17.865	17.904	18.863
	2000	19.363	18.129	17.918	17.945	17.950	18.018	17.862	17.790	17.991	17.879	17.909	18.850
R5	base	19.265	18.090	17.877	17.914	17.915	17.955	17.837	17.827	18.037	17.918	17.905	18.852
	20	19.269	18.096	17.881	17.917	17.921	17.979	17.842	17.832	18.047	17.912	17.913	18.852
	50	19.299	18.100	17.881	17.917	17.926	17.981	17.839	17.832	18.050	17.912	17.915	18.858
	100	19.285	18.101	17.883	17.920	17.927	17.976	17.889	17.846	18.062	17.905	17.916	18.857
	200	19.300	18.107	17.884	17.924	17.930	17.980	17.906	17.848	18.067	17.910	17.921	18.860
	500	19.292	18.106	17.888	17.925	17.934	17.985	17.908	17.852	18.071	17.920	17.923	18.860
	1000	19.302	18.110	17.894	17.934	17.941	17.992	17.917	17.853	18.079	17.922	17.925	18.869
	2000	19.314	18.121	17.904	17.938	17.958	18.009	17.923	17.858	18.087	17.924	17.931	18.873

**Table 4 sensors-23-01298-t004:** Plantar insole— foot size 43 (27 cm).

	Weight	S0	S1	S2	S3	S4	S5	S6	S7	S8	S9	S10	S11
R1	base	10.584	18.271	18.044	18.006	17.948	18.037	17.905	17.876	17.926	18.194	17.954	18.972
	20	10.592	18.274	18.048	18.007	17.951	18.041	17.908	17.880	17.929	18.194	17.957	18.961
	50	10.599	18.277	18.053	18.008	17.959	18.046	17.911	17.884	17.932	18.196	17.961	18.961
	100	10.606	18.279	18.056	18.010	17.963	18.050	17.913	17.888	17.936	18.200	17.966	18.964
	200	10.612	18.283	18.060	18.012	17.965	18.053	17.920	17.893	17.939	18.204	17.973	18.967
	500	10.621	18.288	18.066	18.015	17.970	18.057	17.925	17.900	17.943	18.209	17.979	18.971
	1000	10.632	18.292	18.072	18.020	17.975	18.064	17.929	17.908	17.950	18.217	17.979	18.978
	2000	10.643	18.304	18.081	18.028	17.982	18.073	17.940	17.918	17.960	18.227	17.982	18.983
R2	base	10.627	18.268	18.070	18.008	17.957	18.040	17.911	17.898	17.934	18.189	17.957	18.953
	20	10.631	18.271	18.072	18.009	17.962	18.042	17.916	17.899	17.937	18.190	17.959	18.965
	50	10.633	18.273	18.075	18.010	17.967	18.043	17.919	17.902	17.941	18.191	17.962	18.970
	100	10.636	18.276	18.080	18.012	17.970	18.045	17.925	17.908	17.944	18.195	17.966	18.976
	200	10.640	18.278	18.083	18.015	17.977	18.048	17.929	17.913	17.947	18.198	17.968	18.980
	500	10.649	18.283	18.087	18.018	17.980	18.052	17.934	17.920	17.951	18.204	17.973	18.985
	1000	10.661	18.289	18.091	18.023	17.989	18.058	17.941	17.923	17.957	18.210	17.977	18.988
	2000	10.674	18.296	18.096	18.030	17.996	18.068	17.953	17.929	17.968	18.225	17.994	18.997
R3	base	10.635	18.275	18.065	18.009	17.961	18.023	17.908	17.894	17.925	18.187	17.941	18.932
	20	10.638	18.280	18.066	18.011	17.961	18.033	17.913	17.897	17.928	18.188	17.945	18.941
	50	10.641	18.283	18.070	18.012	17.962	18.035	17.915	17.900	17.929	18.190	17.948	18.946
	100	10.643	18.285	18.073	18.014	17.962	18.039	17.919	17.906	17.931	18.195	17.952	18.951
	200	10.647	18.288	18.076	18.015	17.970	18.041	17.923	17.909	17.932	18.198	17.959	18.952
	500	10.655	18.291	18.079	18.018	17.973	18.045	17.928	17.912	17.936	18.204	17.965	18.961
	1000	10.662	18.297	18.085	18.023	17.987	18.053	17.934	17.918	17.940	18.210	17.968	18.964
	2000	10.688	18.304	18.092	18.030	17.997	18.062	17.946	17.925	17.950	18.222	17.975	18.970
R4	base	10.665	18.284	18.058	18.010	17.965	18.029	17.913	17.882	17.929	18.181	17.947	18.940
	20	10.667	18.286	18.060	18.011	17.969	18.035	17.915	17.885	17.931	18.183	17.950	18.949
	50	10.671	18.288	18.064	18.013	17.974	18.037	17.918	17.888	17.932	18.186	17.952	18.953
	100	10.674	18.290	18.067	18.015	17.977	18.041	17.921	17.893	17.934	18.188	17.957	18.957
	200	10.677	18.292	18.068	18.016	17.980	18.043	17.925	17.896	17.936	18.189	17.960	18.962
	500	10.683	18.297	18.071	18.020	17.983	18.047	17.930	17.899	17.940	18.197	17.964	18.968
	1000	10.693	18.301	18.077	18.025	17.988	18.054	17.938	17.905	17.945	18.204	17.968	18.969
	2000	10.701	18.311	18.087	18.032	17.997	18.064	17.950	17.915	17.954	18.217	17.971	18.978
R5	base	10.617	18.273	18.054	18.014	17.970	18.033	17.922	17.888	17.940	18.184	17.956	18.949
	20	10.620	18.276	18.056	18.016	17.973	18.038	17.925	17.890	17.941	18.186	17.959	18.957
	50	10.624	18.277	18.060	18.017	17.981	18.039	17.927	17.894	17.943	18.189	17.962	18.961
	100	10.627	18.279	18.062	18.019	17.984	18.042	17.932	17.899	17.945	18.190	17.966	18.964
	200	10.631	18.281	18.064	18.020	17.986	18.045	17.936	17.902	17.947	18.192	17.969	18.970
	500	10.639	18.286	18.067	18.023	17.991	18.049	17.941	17.906	17.951	18.197	17.974	18.975
	1000	10.645	18.288	18.071	18.028	17.994	18.058	17.949	17.911	17.957	18.199	17.978	18.980
	2000	10.654	18.296	18.074	18.035	18.004	18.069	17.959	17.919	17.965	18.215	17.984	18.984

**Table 5 sensors-23-01298-t005:** Resistive plantar insole.

	Weight	S0	S1	S2	S3	S4	S5	S6
R1	base	0.003	0.027	0.011	0.024	0.012	0.012	0.013
	20	0.004	0.027	0.010	0.024	0.012	0.012	0.013
	50	0.004	0.027	0.011	0.024	0.014	0.012	0.013
	100	0.004	0.027	0.011	0.024	0.013	0.012	0.013
	200	0.004	0.026	0.011	0.026	0.014	0.012	0.014
	500	0.645	0.243	0.711	0.814	0.222	0.375	0.972
	1000	1.612	1.385	1.738	1.596	1.121	1.395	1.637
	2000	2.341	2.132	2.344	2.209	2.112	2.145	2.141
R2	base	0.003	0.026	0.011	0.025	0.012	0.012	0.013
	20	0.003	0.026	0.011	0.025	0.013	0.012	0.013
	50	0.004	0.026	0.011	0.025	0.014	0.012	0.012
	100	0.004	0.026	0.011	0.025	0.013	0.012	0.012
	200	0.004	0.026	0.011	0.025	0.013	0.012	0.013
	500	0.375	0.911	0.147	0.896	0.280	0.418	0.913
	1000	1.542	1.596	1.680	1.694	1.181	1.335	1.470
	2000	2.269	2.318	2.377	2.306	2.311	2.321	2.021
R3	base	0.004	0.026	0.011	0.025	0.014	0.012	0.013
	20	0.003	0.026	0.011	0.025	0.010	0.012	0.013
	50	0.003	0.026	0.011	0.025	0.014	0.012	0.013
	100	0.003	0.026	0.010	0.025	0.012	0.012	0.013
	200	0.003	0.026	0.011	0.025	0.012	0.012	0.013
	500	0.468	0.633	0.484	0.852	0.136	0.452	1.137
	1000	1.397	1.380	1.519	1.542	1.266	1.536	1.577
	2000	2.340	2.228	2.268	2.320	2.033	2.355	2.281
R4	base	0.004	0.026	0.011	0.024	0.012	0.012	0.013
	20	0.004	0.026	0.011	0.025	0.012	0.013	0.012
	50	0.003	0.026	0.011	0.024	0.012	0.012	0.013
	100	0.003	0.026	0.011	0.025	0.012	0.012	0.013
	200	0.004	0.026	0.011	0.025	0.012	0.012	0.013
	500	0.603	0.793	0.471	1.035	0.309	0.197	0.892
	1000	1.539	1.340	1.026	1.659	1.159	1.654	1.569
	2000	2.137	2.063	2.353	2.229	2.164	2.176	2.179
R5	base	0.003	0.027	0.011	0.025	0.013	0.012	0.012
	20	0.003	0.026	0.011	0.025	0.014	0.012	0.013
	50	0.003	0.027	0.011	0.025	0.012	0.012	0.013
	100	0.004	0.026	0.011	0.025	0.012	0.012	0.013
	200	0.003	0.027	0.012	0.025	0.013	0.012	0.013
	500	0.529	0.779	0.584	1.068	0.268	0.583	0.906
	1000	1.296	1.044	0.910	1.598	1.148	0.914	1.593
	2000	2.148	1.997	1.910	1.975	2.092	2.232	2.260

**Table 6 sensors-23-01298-t006:** Plantar insole—foot size 43 (27 cm)—higher loads.

	Weight (kg)	S0	S1	S2	S3	S4	S5	S6	S7	S8	S9	S10	S11
R1	1000	19.651	18.301	18.077	18.023	17.988	18.054	17.934	17.905	17.948	18.217	17.967	19.020
	2000	19.663	18.311	18.087	18.030	17.997	18.064	17.946	17.915	17.956	18.227	17.975	19.024
	3000	19.672	18.317	18.093	18.034	18.001	18.070	17.952	17.925	17.961	18.233	17.979	19.028
	4000	19.676	18.325	18.100	18.039	18.005	18.076	17.958	17.932	17.965	18.246	17.982	19.032
	5000	19.683	18.330	18.110	18.042	18.009	18.080	17.965	17.938	17.969	18.251	17.986	19.037
	6000	19.689	18.333	18.120	18.046	18.014	18.085	17.968	17.946	17.973	18.255	17.991	19.040
	7000	19.695	18.341	18.129	18.049	18.018	18.089	17.972	17.951	17.976	18.269	17.994	19.044
	8000	19.700	18.346	18.135	18.051	18.020	18.094	17.976	17.956	17.980	18.279	17.998	19.048
R2	1000	19.656	18.306	18.087	18.032	17.994	18.062	17.942	17.915	17.958	18.235	17.976	19.034
	2000	19.668	18.316	18.097	18.040	18.004	18.069	17.953	17.925	17.970	18.252	17.983	19.038
	3000	19.675	18.323	18.107	18.044	18.010	18.078	17.961	17.939	17.976	18.268	17.990	19.043
	4000	19.684	18.335	18.116	18.048	18.016	18.085	17.966	17.944	17.974	18.278	17.993	19.046
	5000	19.692	18.338	18.130	18.052	18.018	18.089	17.971	17.958	17.985	18.299	17.997	19.053
	6000	19.702	18.346	18.143	18.058	18.022	18.098	17.977	17.964	17.992	18.312	18.003	19.057
	7000	19.712	18.355	18.149	18.060	18.026	18.102	17.981	17.971	17.991	18.319	18.005	19.061
	8000	19.718	18.364	18.157	18.063	18.028	18.109	17.987	17.975	17.996	18.328	18.007	19.065
R3	1000	19.659	18.306	18.085	18.029	17.992	18.056	17.939	17.911	17.954	18.210	17.971	19.028
	2000	19.671	18.316	18.091	18.039	18.001	18.063	17.951	17.922	17.965	18.222	17.978	19.034
	3000	19.678	18.324	18.100	18.043	18.006	18.072	17.959	17.934	17.969	18.227	17.985	19.035
	4000	19.682	18.329	18.107	18.045	18.009	18.079	17.964	17.942	17.968	18.235	17.986	19.040
	5000	19.691	18.339	18.119	18.051	18.015	18.083	17.969	17.953	17.975	18.248	17.990	19.047
	6000	19.697	18.342	18.131	18.057	18.019	18.091	17.974	17.960	17.980	18.257	17.994	19.053
	7000	19.704	18.351	18.141	18.057	18.022	18.096	17.978	17.966	17.981	18.269	18.000	19.051
	8000	19.709	18.356	18.148	18.061	18.025	18.101	17.983	17.970	17.986	18.284	18.003	19.057

## Data Availability

Not applicable.

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
