# Peer review of "Smart Shoe Insole Based on Polydimethylsiloxane Composite Capacitive Sensors"

_sensors, 2023, doi:10.3390/s23031298_

Round 1

Reviewer 1 Report

As far as I know, a lot of research using capacitive sensors as a pressure sensing medium have been published so far. However, authors did not cite any of these papers. Therefore readers cannot understand the importance or novelty of this paper.

In actual application of insole for shoes, the insole itself is not only under pressure but also it curves slightly, as the insole is not rigid material. Distortion from the curvature can change the capacitance of the insole. How would you distinguish the effect of pressure from the bottom of the foot and the effect of capacitance change by the curvature?

Reviewer 2 Report

Authors have addressed an interesting research problem and solution, but still major changes are required to be incorporated

Comments for Authors

1. Contribution and significance of the conducted research are unclear so can be rewritten

2. Related work section is missing, so can be created separtely as 'Section 2 Related Work', and for its improvement authors are highly recommended to consider this high quality work as a ref. <An Energy-Efficient Algorithm for Wearable Electrocardiogram Signal Processing in Ubiquitous  Healthcare Applications>

3.     Introduction needs to explain the main contributions of the work clearer.

4.     The novelty of this paper is not clear. The difference between present work and previous Works should be highlighted.

5.     Research gaps, objectives of the proposed work should be clearly justified.

6.     English must be revised throughout the manuscript.

7.     Limitations and Highlights of the proposed methods must be addressed properly

8.     Experimental results are not convincing, so authors must give more results to justify their proposal by comparing with other state of the art existing techniques

Finally, paper needs major changes 

Reviewer 3 Report

This paper proposed an insole for plantar pressure measurement with PDMS. The work covered the concept, fabrication and experiments. However, I think it cannot be accepted at its current status. A major revision is needed.

1. About the sensitivity in table 1. Different thicknesses produce different sensitivities, but the tendency is not monotonic. 4mm has the lowest value, but the others have similar values. Why?Then, why not use 2mm thick scheme, which has the best sensitivity? Moreover, how the test and calculate the sensitivities? How much pressure is loaded?

2. The information in 2.3.1 has been given at the beginning of 2.3. Repeated content is not necessary.

3. It is better to give a label, e.g. a,b,c, for each subfigure in figures for convenience when you refer them.

4. It seems that the sensing units are integrated with the whole insole. Will this influence the performance?

5. A large force/pressure is loaded when the volunteers stand on the insole. However, the loads for performance evaluation ends to 2kg, which much lower than the working load. The results cannot well indicate the real working parameters. A larger load is needed.

6. About the sensitivities in Tables 3&4, and Figures 5,6. The value is not given. The nonlinearity is not referred, which may be more obvious when the load increase.  A larger load is needed. A discussion about the nonlinearity is needed.

7. About the values in Figure 8. What is the reference for normalization? Can you give a pressure distribution map based on these lines? How can you ensure the validity of obtained results without the necessary calibration at the same loads?  

8. Many capacitive sensors for plantar pressure have been reported. The introduction should be improved to cite them and show the real value of your paper.

Round 2

Reviewer 1 Report

Authors corrected the manuscript properly. It should be published as it is.

Reviewer 2 Report

Authors have improved the article, so minor changes are required

1. Limitations of the proposed method must be highlighted in 'conclusion section'

2. Engligh must be proof-read by native speaker

Reviewer 3 Report

The paper has been improved, and I think it can be accepted now.